# Characterization of Inflammasomes and Their Regulation in the Red Fox

**DOI:** 10.3390/ani13243842

**Published:** 2023-12-13

**Authors:** Huijeong Ahn, Dong-Hyuk Jeong, Gilyoung Lee, Suk-Jin Lee, Jeong-Jin Yang, Yo-Han Kim, Tae-Wook Hahn, Sooyoung Choi, Geun-Shik Lee

**Affiliations:** 1College of Veterinary Medicine and Institute of Veterinary Science, Kangwon National University, Chuncheon 24341, Republic of Korea; balloon1981@naver.com (H.A.); lky123001@gmail.com (G.L.); kimyohan@kangwon.ac.kr (Y.-H.K.); twhahn@kangwon.ac.kr (T.-W.H.); choisooyoung@kangwon.ac.kr (S.C.); 2Laboratory of Wildlife Medicine, College of Veterinary Medicine, Chungbuk National University, Cheongju 28644, Republic of Korea; africabear@cbnu.ac.kr; 3National Park Institute of Wildlife Conservation, Gurye 57616, Republic of Korea; mfungus5@knps.or.kr (S.-J.L.); y2j112@knps.or.kr (J.-J.Y.)

**Keywords:** red foxes, *Vulpes vulpes*, inflammasome, cytokine, interleukin-1beta

## Abstract

**Simple Summary:**

Studying inflammasomes in red foxes is crucial for wildlife veterinary medicine, as it helps in treating and preventing inflammatory diseases in foxes. Inflammasomes, protein complexes in innate immune cells, regulate the release of interleukin (IL)-1β, a pro-inflammatory cytokine, in response to cytosolic danger signals, such as pathogens. Limited understanding exists regarding how inflammasome diversity across species influences pathogen carriage in red foxes. Through investigating the activation and intracellular mechanisms of fox inflammasomes, utilizing systems established in humans and mice, we compared the activating mechanism with that of humans and mice. Our research identified both similarities and differences in the fox inflammasome, providing insights that can be leveraged for treating and modulating veterinary medicine in foxes.

**Abstract:**

Background: Inflammasomes recognize endogenous and exogenous danger signals, and subsequently induce the secretion of IL-1β. Studying inflammasomes in the red fox (*Vulpes vulpes*) is crucial for wildlife veterinary medicine, as it can help control inflammatory diseases in foxes. Methods: We investigated the activation and intracellular mechanisms of three inflammasomes (NLRP3, AIM2, and NLRC4) in fox peripheral blood mononuclear cells (PBMCs), using established triggers and inhibitors derived from humans and mice. Results: Fox PBMCs exhibited normal activation and induction of IL-1β secretion in response to representative inflammasome triggers (ATP and nigericin for NLRP3, dsDNA for AIM2, flagellin for NLRC4). Additionally, PBMCs showed normal IL-1β secretion when inoculated with inflammasome-activating bacteria. In inhibitors of the inflammasome signaling pathway, fox inflammasome activation was compared with mouse inflammasomes. MCC950, a selective NLRP3 inhibitor, suppressed the secretion of dsDNA- and flagellin-mediated IL-1β in foxes, unlike mice. Conclusions: These findings suggest that NLRP3 may have a common role in dsDNA- and flagellin-mediated inflammasome activation in the red fox. It implies that this fox inflammasome biology can be applied to the treatment of inflammasome-mediated diseases in the red fox.

## 1. Introduction

Inflammasomes, intracellular multiprotein complexes, belong to pattern recognition receptors (PRRs) and operate within innate immune cells, such as monocytes and macrophages [1,2]. They recognize pathogens and other danger signals and trigger inflammatory responses through the activation of inflammatory caspases [3,4]. Inflammasomes are composed of sensor proteins, adaptor proteins (apoptosis-associated speck-like protein containing a caspase recruitment domain (ASC)), and effector proteins (e.g., inflammatory caspases). Inflammasomes are named based on the sensor protein they utilize [1,2]. The nucleotide-binding oligomerization domain, leucine-rich repeat, and pyrin domain-containing 3 (NLRP3) sensor detects various endogenous and exogenous danger signals, the absent in melanoma 2 (AIM2) and AIM2-like receptors recognize cytoplasmic double-stranded DNA (dsDNA), and the nod-like receptor (NLR) family caspase recruitment domains (CARD) domain-containing protein 4 (NLRC4) specifically detects bacterial proteins (e.g., flagellin), all of which trigger the assembly of their respective inflammasomes: NLRP3, AIM2, and NLRC4 [1]. The activation of inflammasomes plays a crucial role in driving the maturation and secretion of interleukin (IL)-1β and IL-18. Additionally, inflammasomes can induce inflammatory cell death known as pyroptosis by forming pores in the cell membrane, leading to the release of inflammatory mediators [1].

The activation of inflammasomes requires two steps: priming and activation [5]. During the priming step, the expression of genes encoding inflammasome components is upregulated, preparing the cell to form inflammasome complexes [4]. In the activation step, each component binds together to assemble the inflammasome, leading to the activation of caspase-1 and subsequent release of cytokines, as well as the induction of pyroptosis [4]. Lipopolysaccharide (LPS), which is known to induce the priming step, specifically binds to toll-like receptor (TLR) 4, leading to the activation of nuclear factor kappa B (NF-κB) signaling and subsequent expression of target genes [5]. Triggers that induce the activation step of the NLRP3 inflammasome include adenosine triphosphate (ATP), nigericin, and calcium chloride [6,7,8]. Additionally, the NLRC4 and AIM2 inflammasomes are activated by the cytosolic delivery of flagellin and dsDNA [7,8]. Furthermore, inflammasomes selectively operate in response to various bacterial pathogens, such as *Staphylococcus aureus*, *Salmonella* Typhimurium, *Listeria monocytogenes*, and *Escherichia coli* [3]. The mechanism of action of inflammasomes has been predominantly elucidated in the NLRP3 inflammasome [3]. It is known that intracellular K^+^ efflux and the generation of reactive oxygen species (ROS) serve as cellular signaling mechanisms that trigger the assembly of the inflammasome complex [1]. Inflammasomes also play a crucial role in immune responses that drive age-related and metabolic disorders, such as Alzheimer’s disease and diabetes [4,9]. Additionally, inflammasomes have an impact on the prevention and management of infectious diseases [10].

The red fox, *Vulpes vulpes* (Carnivora, Canidae), is a small carnivorous mammal that is widely distributed worldwide. Red foxes were known to inhabit Korea in the past, but became extinct due to indiscriminate hunting and illegal killing [11]. The red fox has been one of the animal species included in the endangered species restoration project in the Republic of Korea since 2011 [11]. Due to this project, red foxes could inhabit human residential areas, making them potential hosts for transmitting various diseases, including rabies, to human populations, similar to raccoon dogs [12]. The study of the inflammasome biology of wildlife including the red fox plays a crucial role in understanding [13]. For instance, bats exhibit a dampened inflammatory response to viral infections due to alterations in their inflammasomes [14,15]. Moreover, ducks possess an active retinoic acid-inducible gene (RIG)-I inflammasome which grants them a heightened resistance against influenza viruses [16,17]. Conversely, chickens lack the RIG-I, rendering them highly susceptible to influenza viruses [16]. Therefore, the current study sought to characterize fox inflammasomes in order to obtain essential insights for the development of treatments for inflammasome-related diseases in veterinary medicine.

## 2. Materials and Methods

### 2.1. Cell Preparation 

Peripheral blood monocytes (PBMCs) were isolated from freshly drawn blood from Korean red foxes (Wildlife Medical Center, Korea National Park Research Institute, Gurye, Republic of Korea) and beagle dogs (Veterinary Medical Teaching Hospital, Chuncheon, Kangwon National University, Republic of Korea) as per the procedure followed in a previous study [7]. Briefly, the blood was drained into heparin tubes and moved to the laboratory. The blood was mixed with an equal volume of phosphate-buffered saline (PBS) in centrifuge tubes (SPL Life Science Co., Pocheon, Gyeonggi-do, Republic of Korea), and then lymphocyte separation medium (LSM; Thermo Fisher Scientific, Waltham, MA, USA) was added into the tube (blood:PBS:LSM = 1:1:1). After centrifuging (500× *g*, room temperature (RT), 30 min), the interphase of the mixture containing monocytes were harvested and placed in a new tube. After centrifuging (500× *g*, RT, 10 min), the remaining cell pellets were added to Red Blood Cell Lysis Buffer (iNtRON Biotech., Sengnam-si, Republic of Korea) and subjected to centrifuging (500× *g*, RT, 10 min). PBMCs were used for experiments immediately on the day they were prepared. Mouse bone marrow-derived macrophages (BMDMs) were differentiated from marrow cells (C57BL/6, Narabio Co., Seoul, Republic of Korea) in media containing 30% L929 cell-conditioned media, 10% fetal bovine serum (FBS), and antibiotics for 7 days as per the procedure followed in the previous study [8]. Animal experiments were conducted in accordance with the Guide for the Care and Use of Laboratory Animals, National Research Council Committee for the Update of the Guide for the Care and Use of Laboratory Animals and were approved by the Institutional Animal Care, and Use Committee of Kangwon National University (IACUC; approval KW-210317-2 for mice, KW-210331-2 for foxes, and KW-210504-1 for dogs). All cells were maintained in Dulbecco’s Modified Eagle Medium (DMEM) or Roswell Park Memorial Institute (RPMI) medium (Welgene Inc., Gyeongsan, Gyeongsangbuk-do, Republic of Korea) containing 10% FBS (Welgene Inc.) and antibiotics (penicillin and streptomycin; Welgene Inc.) at 37 °C in a 5% CO_2_ atmosphere. 

### 2.2. Inflammasome Activation and Inhibition

Cells (2 × 10^6^ cells per well for PBMCs, or 1 × 10^6^ cells per well for BMDMs) were plated in 12-well plates (SPL Life Science), and then treated with lipopolysaccharide (LPS; #L4130, Sigma-Aldrich, St. Louis, MI, USA) in RPMI media containing 10% FBS and antibiotics to induce the priming step. The LPS-primed cells were further treated with the following triggers to activate inflammasome: ATP (InvivoGen, San Diego, CA, USA) for 1 h, nigericin (NG; Tocris Bioscience, Minneapolis, MN, USA) for 1 h, calcium chloride (CaCl_2_; iNtRON Biotech.) for 3 h, dsDNA with jetPRIME™ (Polyplus-transfection Inc., Illkirch, France) for 1 h, flagellin (InvivoGen) with Lipofectamine 2000 (Invitrogen, Carlsbad, CA, USA) for 3 h, and bacterial triggers (*Staphylococcus aureus* for 6 h, *Salmonella* Typhimurium for 1 h, *Listeria monocytogenes* for 3 h, and *Escherichia coli* for 6 h) [1]. The bacteria were cultured at 37 °C on Luria-Bertani media (Condalab, Torrejón de Ardoz, Madrid, Spain) for *S. aureus*, *S.* Typhimurium, *E. coli*, or Brain Heart Infusion media (Condalab) for *L. monocytogenes*. For inflammasome inhibition assessment, the LPS-primed cells were treated with diphenyleneiodonium (DPI, 100 μM, Tocris Bioscience), high potassium solution (KCl, 50 mM; iNtRON Biotech.), glibenclamide (150 μM; Santa Cruz Biotech., Santa Cruz, CA, USA), or MCC950 (200 nM; InvivoGen) in the presence of NG (40 μM) for 1 h, dsDNA (4 μg/mL) with jetPRIME (5 μL/mL) for 1 h, or flagellin (500 ng/mL) with Lipofectamin 2000 (10 μL/mL) for 3 h [18,19,20,21]. After the activation step, the cellular supernatant and lysate were prepared for use in further analyses.

### 2.3. Immunoblotting 

The supernatant and lysate samples were separated with electrophoresis using 10% or 16% sodium dodecyl-sulfate polyacrylamide gel electrophoresis (sodium dodecyl sulfate polyacrylamide gel electrophoresis, [SDS-PAGE]) gels (Mini-PROTEAN^®^ TETRA Handcast system; BIO-RAD, Hercules, CA, USA). Subsequently, the gels were transferred onto a polyvinylidene difluoride membrane (PVDF, Thermo Fisher Scientific). The membranes were blocked with 3% skim milk and probed with anti-dog IL-1β sera (AF3747, R&D Systems, Minneapolis, MN, USA) overnight at 4 °C. In addition, the membrane was probed with horseradish peroxidase (HRP) secondary antibody (Thermo Fisher Scientific) for 1 h and then visualized by applying a chemiluminescence solution (Abfrontier, Seoul, Republic of Korea) and a chemiluminescence imaging system (EZ-Capture II, ATTO Technology, Tokyo, Japan). The membranes were stripped and re-probed with anti-actin sera (#sc-1615, Santa Cruz Biotechnology). 

### 2.4. Assay for IL-1β Secretion

The IL-1β levels in the supernatant were measured by using an enzyme-linked immunosorbent assay (ELISA) kit (DY3747 for foxes and DY401 for mice, R&D Systems). The plates were analyzed using a multi-microplate spectrophotometer (Synergy™ H1 Hybrid Multi-Mode Reader, BioTek, Winooski, VT, USA).

### 2.5. Statistical Analysis

Statistical software (GraphPad Prism 6, San Diego, CA, USA) was used to perform analyses using the Mann–Whitney or Kruskal-Wallis tests. The *p*-value is shown in the figures.

## 3. Results

### 3.1. Optimization of Inflammasome Priming and Activation in Fox PBMCs

The gene expression of inflammasome components and inflammatory cytokines in red foxes was analyzed using specific primers designed for dogs due to limited information on the fox genes (Appendix A). Although the gene sequences of red foxes and dogs are not identical (Appendix A), it was confirmed that the dog primers used are functional in red foxes as well. Upon treatment with LPS, both dog and fox PBMCs exhibited increased expression of inflammasome component genes (pro-IL-1β and NLRP3) and inflammatory cytokine genes (IL-1α, tumor necrosis factor (TNF)α, IL-6, and IL-10) except for ASC. It indicates that LPS induces gene expression through the TLR4/NF-κB signaling pathway in fox PBMCs, suggesting that LPS can serve as an inducer for the priming step in fox PBMCs, similar to other animals. Moreover, these findings imply that antibodies designed for dogs can be used to assess inflammasome activation in red foxes.

Both dog and fox PBMCs were primed with LPS and, subsequently, their activation step was induced by inflammasome triggers, such as NG and ATP (Figure 1A). As shown in Figure 1B, LPS-primed cells of dogs and red foxes exhibited the expression of pro-IL-1β protein in the cellular lysate. The pro-IL-1β (around 31 kDa) was cleaved into IL-1β (p17) upon the NG- and ATP-mediated inflammasome activation, and subsequently released from the cells into the cellular supernatant. This indicates that dog IL-1β antibodies can detect the secreted fox IL-1β (Appendix A), a readout of inflammasome activation. Furthermore, it was revealed that the known triggers of NLRP3 inflammasome, namely NG and ATP, can induce activation of inflammasomes in red foxes.

### 3.2. NLRP3 Inflammasome Triggers in Fox PBMCs

To optimize the priming step, which is crucial for NLRP3 inflammasome activation [5], fox PBMCs were treated with various concentrations of LPS in the presence or absence of NLRP3 triggers (ATP and NG). Treatment with LPS in a concentration-dependent manner led to a small but detectable secretion of IL-1β through LPS priming alone (Figure 2A). However, no significant increase in IL-1β secretion was observed. Furthermore, there were no significant differences in IL-1β secretion when ATP and NG were administered across different LPS concentrations. ELISA-based studies consistently demonstrated the selective detection of cleaved fox IL-1β (p17) in the cellular supernatant using the dog IL-1β ELISA kit (Appendix A). Therefore, priming with 1 ng/mL LPS ensures robust NLRP3 inflammasome activation in fox PBMCs. Subsequently, the concentration-dependent effects of NLRP3 triggers, including ATP, NG, and CaCl_2_, were evaluated for NLRP3 inflammasome activation in fox PBMCs (Figure 2B). The results revealed that all three NLRP3 triggers significantly induced the secretion of IL-1β (p17) in a concentration-dependent manner. In conclusion, the NLRP3 inflammasome is effectively activated by the NLRP3 triggers in red foxes.

### 3.3. AIM2 and NLRC4 Inflammasome Triggers in Fox PBMCs

The NLRP3 inflammasome can recognize both danger- and pathogen-associated molecular patterns (DAMPs and PAMPs), while the AIM2 and NLRC4 inflammasomes specifically specialize in detecting PAMPs [1]. To activate the fox AIM2 inflammasome, cytoplasmic dsDNA was introduced into the cells. Similarly, the fox NLRC4 inflammasome was activated by delivering flagellin into the cytoplasm. The optimal concentration of LPS for priming the AIM2 inflammasome in red foxes was determined by evaluating various concentrations of LPS, and it was found that 1 ng/mL of LPS priming resulted in the optimal secretion of IL-1β (Figure 3A and Appendix A). In LPS-primed fox PBMCs, the introduction of dsDNA into the cytoplasm led to a concentration-dependent secretion of IL-1β (Figure 3B). Similarly, the appropriate LPS priming concentration for fox NLRC4 inflammasome activation was determined to be 1 ng/mL, which induced optimal secretion of IL-1β (Figure 3C). As shown in Figure 3D, LPS-primed fox PBMCs treated with various concentrations of flagellin introduced into the cytoplasm exhibited a concentration-dependent secretion of IL-1β. In conclusion, dsDNA and flagellin, the well-known triggers of AIM2 and NLRC4, might be effective in activating the AIM2 and NLRC4 inflammasomes in red foxes. 

### 3.4. Bacterial Inflammasome Triggers in Fox PBMCs

The host detects pathogen invasion through cytosolic PRRs such as NLRP3, AIM2, and NLRC4, and induces immune responses to resistant infection through the inflammasome assembly [3]. Various types of inflammasomes have been identified that are activated depending on the type of pathogen [1,4]. In this study, the LPS-primed fox PBMCs and mouse BMDMs were inoculated with the well-characterized bacteria (i.e., *Staphylococcus aureus*, *Salmonella* Typhimurium, *Listeria monocytogenes*, and *Escherichia coli*) to trigger the inflammasome activation. These bacteria can simultaneously stimulate more than one type of inflammasome [3]. The secretion of IL-1β was measured as an indicator of inflammasome activation (Figure 4). All the pathogens induced IL-1β secretion from fox PBMCs in a dose-dependent manner, comparable to that from mouse BMDMs (Appendix A). Taken together, it was found that the bacterial triggers, known for inflammasome activation in humans and mice, were capable of activating inflammasomes in fox cells.

### 3.5. Mechanistic Studies of Fox Inflammasomes

Inflammasome sensors directly or indirectly recognize danger signals (e.g., ROS or K^+^ efflux) or danger from specific factors (e.g., dsDNA or flagellin) in the cytoplasm, and initiate the assembly of the inflammasomes [1,2]. To investigate the cytosolic signaling involved in fox inflammasome activation, representative inhibitors of inflammasome activation were utilized to treat fox PBMCs during the activation step (Figure 5A). The results showed that NG- and dsDNA-mediated IL-β secretion in fox PBMCs was inhibited when treated with a scavenger of ROS (DPI), inhibitors of potassium efflux (KCl and glibenclamide), and a selective inhibitor of NLRP3 inflammasome assembly (MCC950). Conversely, flagellin-mediated IL-1β secretion in fox PBMCs was suppressed by DPI, KCl, and MCC950, while no significant change was observed with glibenclamide. The results (Figure 5B) showed that all inhibitors suppressed the activation of NG-mediated IL-1β releases in both foxes and mice. However, glibenclamide and MCC950 were unable to inhibit IL-1β secretion in murine macrophages when transfected with dsDNA and flagellin. Interestingly, IL-1β secretion in response to dsDNA and flagellin transfection in red foxes was found to be suppressed by MCC950, which is known as a selective inhibitor of NLRP3 inflammasome in humans and mice [22]. Overall, the NLRP3 inflammasome shares the same intracellular signaling pathway in both mice and red foxes, but the pathway of inflammasome activation in response to dsDNA and flagellin in red foxes may operate differently from that in mice.

## 4. Discussion

In this study, the response of fox PBMCs to LPS priming and inflammasome triggers for the priming and activation steps was similar to that of other species [5,7,23,24]. However, differences were observed in inhibitor studies (Figure 5). Both fox and mouse NLRP3 showed similar responses to inhibitors. Specifically, the ROS scavenger (DPI), K^+^ efflux inhibitors (KCl and glibenclamide), and NLRP3 selective inhibitor (MCC950) suppressed IL-1β secretion induced by NG-mediated NLRP3 inflammasome activation in both foxes and mice. These intracellular molecular pathways have been validated as NLRP3 inflammasome activation pathways in humans and mice [2]. However, for the AIM2 inflammasome triggered by dsDNA transfection, the results observed for red foxes and mice were different. In red foxes, all the inhibitors (DPI, KCl, glibenclamide, and MCC950) suppressed dsDNA-induced IL-1β secretion, whereas, in mice, suppression was only observed with DPI and KCl. Both KCl and glibenclamide inhibit K^+^ efflux, the upstream pathway of NLRP3 inflammasome activation [2], but a high concentration of KCl inhibits it via high hydrostatic pressure [18], while glibenclamide inhibits it via ATP-sensitive potassium channels [19]. Therefore, glibenclamide is selective for NLRP3, while KCl inhibits the AIM2 inflammasome [18,19]. MCC950, known as a selective NLRP3 inhibitor [20,22], directly inhibits NLRP3 but does not inhibit IL-1β secretion induced by the AIM2 inflammasome in mice. However, MCC950 inhibited IL-1β secretion induced by dsDNA transfection in red foxes in the current study. Regarding NLRC4 inflammasome activation triggered by flagellin, DPI, KCl, and MCC950 inhibited IL-1β secretion in red foxes. In mice, DPI inhibited IL-1β secretion mediated by flagellin, and partial inhibition was observed with high-concentration KCl as well. Glibenclamide had no impact on IL-1β secretion induced by NLRC4 inflammasome in both mice and red foxes. Interestingly, MCC950 did not affect mouse NLRC4 inflammasome activation, but it inhibited IL-1β secretion induced by flagellin in foxes. That is, MCC950 inhibited not only dsDNA- and flagellin-mediated inflammasome activation, but also NLRP3 inflammasome in red foxes. MCC950 inhibits the NLRP3 inflammasome by directly targeting the NLRP3 NACHT domain and interfering with the Walker B motif function, preventing conformational changes and oligomerization of NLRP3 [20]. Previous studies have reported that MCC950 does not affect AIM2 and NLRC4 inflammasome activation [22]. Gene deficiencies in AIM2 and NLRC4 may be seen in various animals other than humans and mice [13,25]. These imply that alternative proteins in red foxes compensate for the loss of AIM2 and NLRC4, the cytosolic sensors of dsDNA and flagellin, and collectively activate the NLRP3 inflammasome through a unique mechanism. We used transfection of dsDNA and flagellin to selectively activate AIM2 and NLRC4 inflammasomes in the current study. However, MCC950 suppressed dsDNA- and flagellin-mediated IL-1β secretion in fox PBMCs. Cytosolic dsDNA activates the cell death programs through the cGAS-STING axis pathway and increases intracellular K^+^ efflux to activate NLRP3 inflammasome [26,27]. Furthermore, flagellin transfection activates NLRP3 inflammasome through ROS and cathepsin-dependent mechanisms [28]. LPS also enters the cytoplasm and activates NLRP3 inflammasome through caspase-4/5 (caspase-11 for mice) activation, and IL-1β secretion is inhibited by MCC950 [29,30]. Based on this, we hypothesized that fox inflammasome activation shares a common NLRP3 inflammasome pathway with mice and humans, while having different intracellular signaling pathways for dsDNA- and flagellin-mediated inflammasome activation compared to the known pathways in humans and mice. 

Genomic studies have been conducted to investigate the expression and function of the inflammasome components in various animal species, including red foxes [13,31]. It was found that the AIM2 gene is not present in red foxes [13,25]. AIM2 is a sensor protein that recognizes intracellular dsDNA, and AIM2-like receptors (i.e., myeloid cell nuclear differentiation antigen (MNDA), pyrin and HIN domain family member 1 (PYHIN1), and interferon-gamma-inducible protein 16 [IFI16]) with similar functions exist [31]. Therefore, the absence of AIM2 in red foxes does not imply the lack of functioning AIM2 inflammasomes. This study revealed that red foxes possess an AIM2-like receptor that can sense the presence of cytosolic dsDNA and activate the inflammasome, leading to the secretion of mature IL-1β. Similar results were obtained in a previous study on porcine AIM2 inflammasome [23]. Like red foxes, pigs also lack the AIM2 gene, but the AIM2 inflammasome can be activated through dsDNA transfection [23]. In porcine PBMCs, treatment with KCl and glibenclamide did not inhibit AIM2 inflammasome activation, but DPI did [23]. These results indicate that the regulation of the AIM2 inflammasome may vary among different animal species. In addition to AIM2, the NLRC4 and NLR family of apoptosis inhibitory proteins (NAIP) genes are also not expressed in red foxes [13]. The NLRC4 inflammasome is assembled upon the detection of intracellular flagellin by NAIP (e.g., Naip 5 and 6) [1]. Therefore, NLRC4 and NAIP are essential components for NLRC4 inflammasome activation. However, the current study successfully induced IL-1β secretion through the cellular uptake of flagellin in red foxes (Figure 3). *Salmonella* Typhimurium, a representative trigger for NLRC4 inflammasome [32], also successfully induced IL-1β secretion in fox PBMCs. Thus, there may be alternative sensors in the red fox cells that can substitute for NLRC4/NAIP. In addition, inhibitor studies showed that KCl and MCC950 inhibited fox NLRC4 inflammasomes (Figure 5), suggesting the involvement of a different form of inflammasome activation, rather than the canonical NLRC4 inflammasome. In particular, based on this study, it is anticipated that the NLRP3 inflammasome may play a common role in the activation of both AIM2 and NLRC4 inflammasomes in foxes, as evidenced by the inhibitory effects of the NLRP3 selective inhibitor, MCC950, on dsDNA- and flagellin-mediated IL-1β secretion in red foxes.

## 5. Conclusions

In this study, we investigated the activation of three representative inflammasomes (i.e., NLRP3, AIM2, and NLRC4) and their molecular regulatory mechanisms in red foxes. Red foxes are members of the Canidae family and share a similar nucleotide sequence and amino acid sequence of pro-IL-1β with dogs. Through this study, we have revealed that the use of antibodies and primers derived from dogs can be applied to fox inflammasome research. Additionally, it has been observed that fox PBMCs secrete mature IL-1β (p17) in response to ATP, NG, and CaCl_2_, which are typical triggers of NLRP3 inflammasome activation [6,7,8]. Additionally, we found that fox PBMCs increased secretion of IL-1β (p17) upon the cytosolic introduction of dsDNA and flagellin, which are triggers for AIM2 and NLRC4 inflammasomes, respectively [18,32]. To induce the activation of inflammasomes, we used LPS, a well-known inducer of inflammasome priming [5]. Fox PBMCs were capable of completing the inflammasome priming process through the LPS/TLR4 signaling pathway, similar to other species [5,7,24]. Moreover, fox PBMCs increased IL-1β secretion in response to inflammasome-inducing bacteria [1]. Although the degree of IL-1β secretion by inflammasome activation differed between fox PBMCs and mouse BMDMs, the pattern of IL-1β secretion induced by known inflammasome triggers was similar to mice. This indicates that the representative inflammasome triggers in red foxes may be presumed to function normally. To investigate the intracellular signaling pathways of fox inflammasomes, we used known inflammasome inhibitors [18,19,20,21]. The results revealed that the fox NLRP3 inflammasome was inhibited by the inhibitors similar to humans and mice, but there were differences observed for dsDNA- and flagellin-mediated inflammasome activation in foxes. Therefore, red foxes appear to have different inflammasome signaling pathways for cytosolic dsDNA and flagellin compared to the pathways known in humans and mice. These findings provide valuable information on inflammasome biology in red foxes. 

## Figures and Tables

**Figure 1 animals-13-03842-f001:**
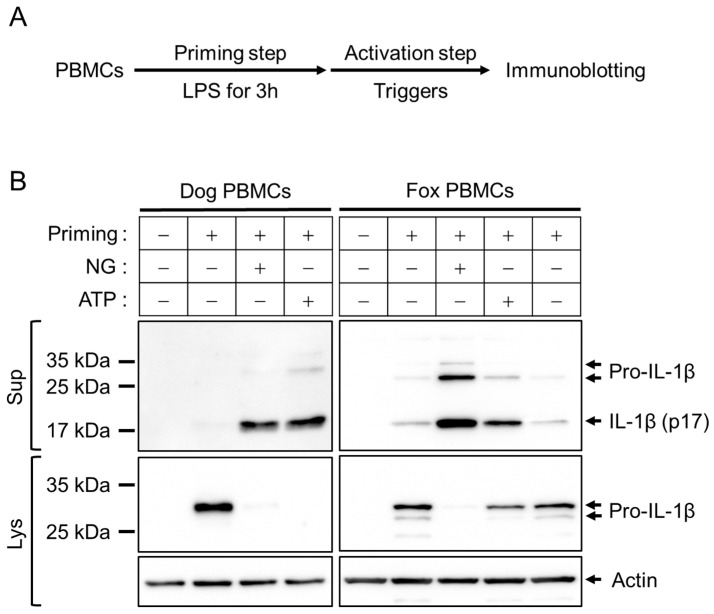
Comparison of inflammasome activation between dog and fox PBMCs. (**A**) Schematic diagram representing the priming and activation steps of inflammasomes. (**B**) LPS-primed PBMCs were treated with NG (40 μM) and ATP (5 mM) for 1 h, and the expression of pro-IL-1β in the cellular lysate (Lys) and the secretion of IL-1β (p17) in the cellular supernatant (Sup) were detected by immunoblotting. Lys: cellular lysate; Sup: cellular supernatant.

**Figure 2 animals-13-03842-f002:**
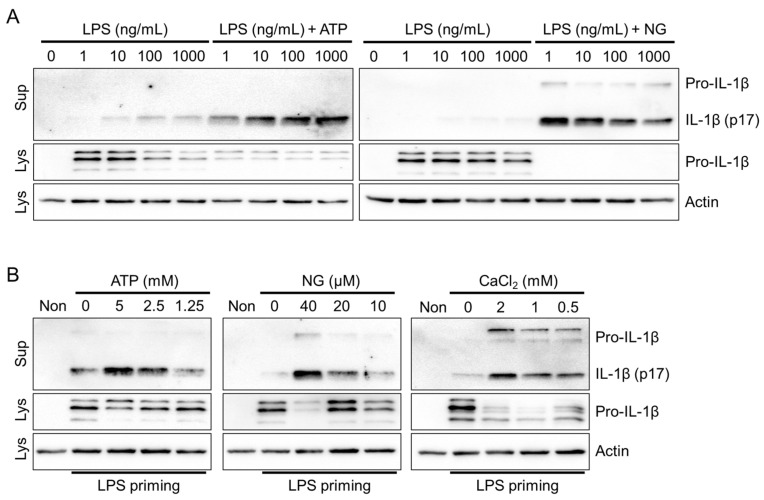
Optimization of the NLRP3 inflammasome in red foxes. (**A**) Fox PBMCs were primed with LPS (1 to 1000 ng/mL) and further treated with ATP (5 mM) or NG (40 μM). (**B**) Fox PBMCs primed with LPS (1 ng/mL) were treated with ATP, NG, and CaCl_2_ at indicated concentrations. The secreted IL-1β was analyzed by immunoblotting. Images are representative of two independent experiments. Lys: cellular lysate; Sup: cellular supernatant.

**Figure 3 animals-13-03842-f003:**
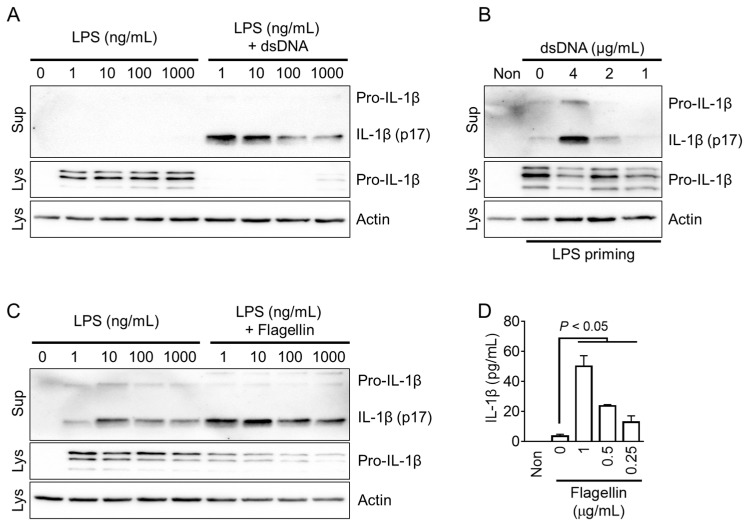
Optimization of AIM2 and NLRC4 inflammasomes in red foxes. (**A**) Fox PBMCs were primed with LPS (1 to 1000 ng/mL) and further transfected with dsDNA (4 μg/mL). (**B**) LPS (1 ng/mL)-primed fox PBMCs were transfected with dsDNA at indicated concentrations. (**C**) Fox PBMCs were treated with LPS (1 to 1000 ng/mL), followed by the introduction of flagellin (500 ng/mL). (**D**) LPS-primed PBMCs of red foxes were treated with flagellin as presented. IL-1β secretion was measured using immunoblotting or ELISA. Images are representative of two independent experiments, and bar graphs present the mean ± standard deviation (SD) of at least three independent experiments. Lys: cellular lysate; Sup: cellular supernatant.

**Figure 4 animals-13-03842-f004:**
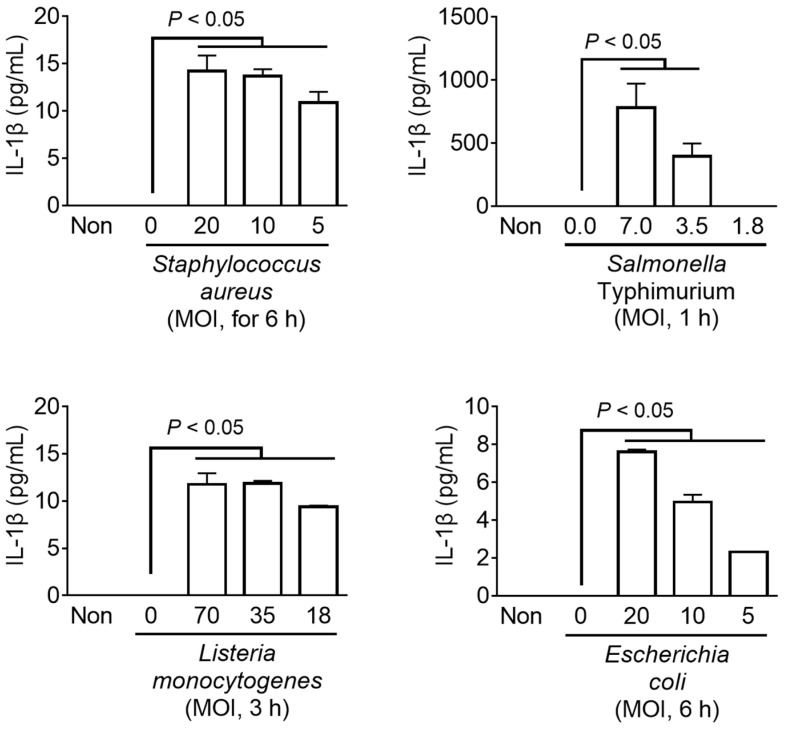
Effect of bacteria on inflammasome activation in fox PBMCs. Fox PBMCs were primed with LPS (1 ng/mL) and then inoculated with bacteria at the indicated multiplicity of infection (MOI) for the presented times. The release of IL-1β was measured using ELISA. The bar graphs present the mean ± SD of at least three independent experiments.

**Figure 5 animals-13-03842-f005:**
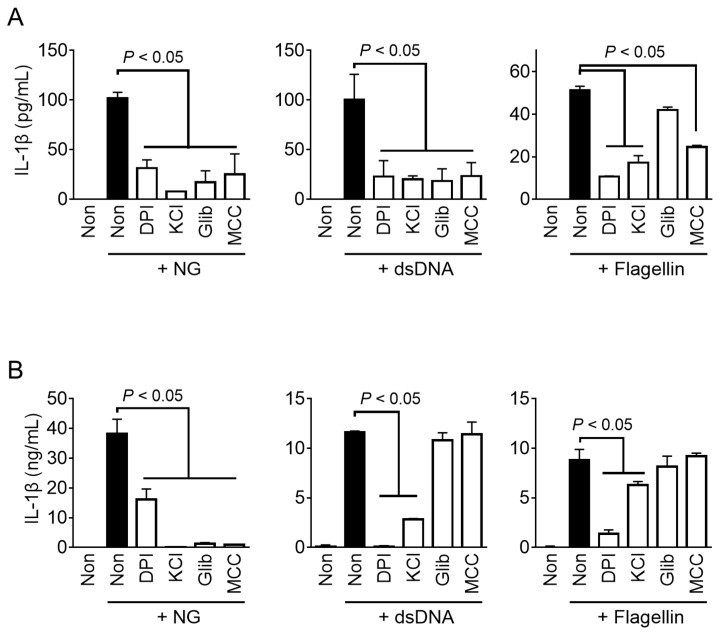
Effect of inhibitors on the activation of fox and mouse inflammasomes. LPS-primed fox PBMCs (**A**) and mouse BMDMs (**B**) were treated with a ROS scavenger (diphenyleneiodonium (DPI)), a potassium efflux inhibitor (KCl and glibenclamide (Glib)), and an NLRP3 selective inhibitor (MCC950 (MCC)) in the presence of inflammasome triggers such as NG for NLRP3, dsDNA for AIM2, and flagellin for NLRC4. The secretion of IL-1β was analyzed by ELISA. Bar graphs present the mean ± SD of at least three independent experiments.

## Data Availability

Data are contained within the article.

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
