# Peer review of "Characterization of Inflammasomes and Their Regulation in the Red Fox"

_animals, 2023, doi:10.3390/ani13243842_

Round 1
Reviewer 1 Report
Comments and Suggestions for Authors
Manuscript ID: animals-2768120
Review report on the article entitled “Characterization of inflammasomes and their regulation in the red fox”
The aim of the study was to investigate NLRP3, AIM2 and NLRC4 inflammasome biology in red fox PBMC. The authors have demonstrated that dog antibodies and primers can be used for fox inflammasome research. The lack of supportive evidence and the interpretation of the AIM2 and NLRC4 inflammasome results (fox PBMC) are problematic and it cannot be published without major changes.
Major concerns
Only evidence for NLRP3 was provided (Supplementary Figure S1.). The authors assumed that dsDNA activated the AIM2 inflammasome and flagellin activated the NLRC4 inflammasome solely based on the detection of IL-1β and because these ligands are known activators of these inflammasomes. However, studies have also demonstrated that flagellin can activate NLRP3 and that cGAS-STING signaling (the major dsDNA sensing pathway) can lead to NLRP3 activation during cellular stress conditions. Furthermore, the authors have not taken the impact of using LPS into consideration: Cytoplasmic LPS activates the CASP4-dependent non-canonical inflammasome. Activated CASP4 induces pyroptosis. Additionally, CASP4-induced pyroptosis and subsequent potassium efflux after permeabilization of the plasma membrane by GSDMD results in the activation the canonical NLRP3 inflammasome. The fact that the secretion of IL-1β was detected via LPS priming alone indicated that the NLRP3 inflammasome was activated in all LPS primed fox PBMC samples. It seems that ATP, nigericin, flagellin and dsDNA via either direct or indirect mechanisms reinforced the NLRP3 inflammasome, resulting in increased IL-1β secretion. This was further demonstrated with MCC950, the selective NLRP3 inhibitor, which suppressed IL-1β secretion in all fox PBMC samples.
There are no results that showed if the AIM2 and NLRC4 (or -like) inflammasomes were even present or activated in the first place. All the statements/speculations (e.g. Lines 245-248 In conclusion, dsDNA and flagellin, the well-known triggers of AIM2 and NLRC4, were effective in activating the AIM2 and NLRC4 inflammasomes in red foxes) about the AIM2 and NLRC4 inflammasomes are based on presumptions.
It is unacceptable to attribute results to an inflammasome without clearly demonstrating that the inflammasome was present in that sample. To conclusively associate results with a specific inflammasome, the same sample should show the presence of that inflammasome as well as the absence the other inflammasomes.
The protein results for AIM2 (or -like) and NLRC4 (or -like) must be shown. Additionally, include the protein results for NLRP3, CASP1, CASP4 (or equivalent in red foxes) and GSDMD.
Specific comments:
Lines 17-19, Lines 25-26. Replace pro-inflammatory cytokines with IL-1β and IL 18, here as well as throughout the manuscript. Activated CASP1 induces the processing, maturation and secretion of IL-1β and IL 18 only.
Lines 17-19, Lines 45-47. Delete cytosolic and cytoplasmic. Many PRRs [that can function as signal 1 (priming) for Inflammasomes] are expressed on the cell surface or in endosomes. Also, correct this throughout the manuscript.
Lines 45-47: ‘Inflammasomes are multiprotein complexes that are assembled by cytoplasmic pattern recognition receptors (PRRs) in innate immune cells, such as monocytes and macrophages, responsible for innate immunity.’ - This is incorrect. Inflammasomes are not physically assembled by cytoplasmic PRRs. Rephrase and rewrite. Additionally, delete ‘responsible for innate immunity’.
Comments on the Quality of English Language
minor editing required
Reviewer 2 Report
Comments and Suggestions for Authors
The original article under the title "Characterization of inflammasomes and their regulation in the red fox" was submitted by Huijeong et al., for publication in Animals. The study focuses on red fox inflammosome activation, which is quite known in many species, including cornivores. However, inflammosome activation and its downward signalling can uncover disease control strategies in red foxes. The authors stated that it can help control or treat zoonotic diseases. Authors need to explain how zoonotic diseases can be treated while humans have different physiologies of infalmmosomes. Or authors can omit the zoonotic disease treatment and keep the control strategies. Further authors have to prove or explain the red fox zoonotic transmission vessels and how inflammosomes can play a role to breaking the zonotic chennel in the discussion and conclusion parts. Yes, this can be helpful for red fox disease prevention. Italicize the bacterial species name throughout the manuscript. Check for typing and grammar errors. Authors have to be limited with inflammosome physiology in foxes or have to privde stong experimental and discussion bassed evidences for zoonotic controls.
Comments on the Quality of English LanguageTyping or minor grammar check
Round 2
Reviewer 1 Report
Comments and Suggestions for Authors
Manuscript ID: animals-2768120
Review report 2 on the article entitled “Characterization of inflammasomes and their regulation in the red fox”
The authors have made the necessary changes to not associate the results to the AIM2 and NLRC4 inflammasomes.
Comments on the Quality of English Language
minor editing